# Clinical Utility of Three-Dimensional Echocardiography in the Evaluation of Mitral Valve Disease: Tips and Tricks

**DOI:** 10.3390/jcm12072522

**Published:** 2023-03-27

**Authors:** Paolo G. Pino, Andrea Madeo, Fabiana Lucà, Roberto Ceravolo, Stefania Angela di Fusco, Francesco Antonio Benedetto, Giovanni Bisignani, Fabrizio Oliva, Furio Colivicchi, Michele Massimo Gulizia, Sandro Gelsomino

**Affiliations:** 1Former Cardiology Department, San Camillo Forlanini Hospital, 00151 Roma, Italy; 2Cardiology Department, Ferrari Hospital, 87012 Castrovillari, Italy; 3Cardiology Department, Grande Ospedale Metropolitano, GOM, AO Bianchi Melacrino Morelli, 89129 Reggio Calabria, Italy; 4Cardiology Unit, Giovanni Paolo II Hospital, 88046 Lamezia, Italy; 5Cardiology Department, San Filippo Neri Hospital, 00135 Rome, Italy; 6De Gasperis Cardio Center, Niguarda Hospital, 20162 Milan, Italy; 7Cardiology Complex Unit, Garibaldi Nesima Hospital, 95122 Catania, Italy; 8Cardiothoracic Department, Maastricht University, 6211 LK Maastrich, The Netherlands

**Keywords:** mitral valve disease, echocardiography, three-dimensional

## Abstract

Although real-time 3D echocardiography (RT3DE) has only been introduced in the last decades, its use still needs to be improved since it is a time-consuming and operator-dependent technique and acquiring a good quality data can be difficult. Moreover, the additive value of this important diagnostic tool still needs to be wholly appreciated in clinical practice. This review aims at explaining how, why, and when performing RT3DE is useful in clinical practice.

## 1. Introduction

Real-time three-dimensional echocardiography (RT3DE) is a new valuable imaging tool for studying the mitral valve (MV) anatomy in each plane orientation [1,2]. Mitral regurgitation (MR) is one of the most common valve disease, and its prevalence increases with age [3].

Recent developments in this technique allow to assess the detailed anatomical characterization of the MV morphology, the precise quantification of the regurgitation and the mechanisms of MV disease (MVD). The most appropriate treatment and the optimal timing can be determined and this technique has assumed a crucial role in MR management [1,3,4,5,6]. However, RT3DE is still not widespread used in clinical practice since it is still not well understood [7]. This review aims to summarize the current knowledge about RT3DE, providing technical and practical information to facilitate the RT3DE data set acquisition.

## 2. Know How

Tridimensional transthoracic echocardiography (3D-TTE) can be performed either from the parasternal or apical approach [8,9]; it is a helpful tool for diagnosis, although it provides less detailed information than three-dimensional transesophageal echocardiography (3D-TEE) (Figure 1). 3D-TEE is performed through the mid-esophageal view at 0° (4 chambers), 90° (2 chambers), and 120° (aortic valve long axis view) [8,9] and provides unique RT3DE images [9,10,11].

### 2.1. Data Acquisition

By simultaneous multiplanar mode (X-plane) two-dimensional (2D) imaging, two orthogonal planes can be simultaneously visualized. The X-plane 2D imaging allows both an excellent spatial and temporal resolution, resulting in an optimal study of challenging findings such as ruptured chordae or vegetations (Figure 2A). The X-plane views involve a more accurate spatial visualization of the MV apparatus, allowing an optimal RT3DE acquisition. The MV volumetric data set is obtained by three main modalities:(1)Live RT3DE mode: Characterized by an optimal temporal resolution despite the low spatial resolution. This modality is useful to test the gain optimization.(2)Zoom mode: If zoom RT3DE is activated, two orthogonal 2D preview images show the ‘‘truncated’’ pyramid sector that the operator can move over the region of interest, regulating width and length and preventing respiratory or rhythm artifacts. However, it is limited by a low temporal resolution that can be improved by minimizing the sector’s width and length. (Figure 2B).(3)“Wide angle single beat”: This option has the same advantages of the zoom mode but a relatively lower temporal and spatial resolution.(4)“Full volume multiple beat” with or without color Doppler. This modality implies an electrocardiogram (ECG) synchronization permitting the analysis of an extensive volumetric data set by acquiring narrow subvolumes (two-seven sequential cardiac cycles). High spatial and temporal resolutions and a global MV view with excellent image quality represent the greatest strength of this modality (Figure 2C).

Among these three modalities, only the “Full Volume” allows the RT3DE representation of the color Doppler (Figure 2D) to be achievable in the same acquisition process. RT3DE color is recommended for estimating MR [12,13].

The full volume modality is the gold standard for anatomical and functional characterization of MV by RT3DE. However, breathing, patient movement, and arrhythmias limit this modality. We recommend applying some practical tricks to avoid these artifacts.

The full volume modality is not provided with the color acquisition. The volume data set is acquired during a single heartbeat. The multi-beat acquisition (full volume) during a single breath-hold allows to post-process volume data set and RT3DE rendering. RT3DE is particularly useful in identifying the main regurgitant jets in the pre-operative planning of MV transcatheter edge-to-edge repair (MV-TEER). Notably, on one hand RT3DE is limited by lower frame rates, but on the other hand full volume modality is affected by stitching artifacts.

### 2.2. Mitral Valve Quantification (MVQ) and Mitral Valve Navigation (MVN)

The method allows the MV apparatus to be rendered and quantified in more than 50 measures around the annulus, leaflets, coaptation points, and tendinous chordae can be challenging.

Mitral valve quantification and mitral valve navigation (MVQ/MVN) analysis can be performed only by 3D-TTE using dedicated software. We suggest using a checklist for an optimal acquisition:

Step 1. The full volume or zoom RT3DE image is obtained, including the entire mitral annulus, the papillary muscle tips, and the left ventricle outflow tract (LVOT) with aortic root.

Step 2. The dataset acquired is uploaded on the workstation, and the best RT3DE images are selected to run in the MV navigation tool.

Step 3. The RT3DE image is split into three orthogonal spatial planes like multiplanar reconstruction (two longitudinal planes and one transversal plane). A fourth box shows a navigation guide (tutorial box).

Step 4. The end-systolic frame position is automatically detected by the software or manually selected by the operator.

Step 5. The planes are aligned following the schematic illustration of the tutorial box.

Step 6. The operator selects six landmarks: four annular points (anterolateral, posteromedial, anterior, and posterior), the nadir of the leaflets, and the aortic point to allow an automatic RT3DE reconstruction of the mitral annulus.

Step 7. Annulus editing is useful for checking the correct positioning of the other annular points by sliding all the longitudinal slices. If no editing is necessary, proceed to the next step. At the end of this step, you can see an RT3DE annulus model overlapped on the RT3DE image.

Step 8. Commissure editing: the correct commissural position is confirmed by comparing it with an RT3DE image box.

Step 9. Leaflet editing: leaflets are automatically detected and drawn in their extension. Manual repositioning of the leaflet points is possible. The coaptation point is manually selected by the operator.

Step 10. Border editing: scallop and commissure border points can be manually adjusted.

Step 11. The papillary muscle tips must be marked to determine chord length if visible.

Based on the reconstructed MV model, a lot of normal and pathological anatomical parameters are automatically calculated for the components of the MV apparatus, such as annular diameters, height, leaflet area, surface area, leaflet billowing height and volume, tenting height and volume, and leaflet closing angles, etc., (Figure 3).

### 2.3. Tips and Tricks for RT3DE Optimization

For obtaining an optimal RT3DE image quality, three main rules should be followed:

(1) “Suboptimal 2D images result in suboptimal RT3DE data sets”. Therefore, optimizing pre-acquisition 2D gains is crucial;

(2) Respiratory and rhythm artifacts must be avoided or reduced;

(3) Acquisition modality choice should vary according to the information that is needed.

We suggest verifying the RT3DE imaging quality by activating the RT3DE live modality. Volumetric RT3DE image quality strictly depends on pre-acquisition 2D gain, so we usually set the gain in a mid-range (no greater than 60) and the compression/dynamic range in a mid-range (Figure 4).

To reduce respiratory and cardiac artifacts, the patient should be aware that the procedure is not dangerous and collaborate during the exam (i.e., stop breathing for a few seconds). Pharmacological sedation is performed with an intravenous benzodiazepine such as midazolam or with propofol (if it is not contraindicated).

We could use a low B-blocker dose (metoprolol 1-2 mg i.v.) to reduce heart rate (HR) and minimize HR disturbance. An HR ranging from 70 to 85 bpm allows an increased number of scan lines per second, optimizing the zoom RT3DE and full volume acquisition modalities.

We use the full volume modality providing that it is suitable. A broader sector acquisition is obtained by maintaining a higher frame rate and, consequently, a good quality RT3DE image.

We switch to the RT3DE zoom modality if the patient is not compliant and if the MVD does not require a large sector or a high frame rate. Highly moving vegetation are better detected by full volume, whereas RT3DE zoom may successfully study a prolapsing scallop.

### 2.4. Post-Processing (Multiplanar Reconstruction and Cropping)

Multiplanar reconstruction (MPR) is essential in RT3DE analysis. The manual rotation of the dataset around a center point allows proper visualization [14,15].

MPR allows a more accurate measurement of the MV area, annulus, vena contracta area, vegetations, and masses by choosing a better section plane. This phase of reconstruction depends on the final image quality.

We prefer to use multiplanar reconstruction before the cropping. By moving the three planes (longitudinal, transversal, and sagittal) perpendicularly at the region of interest, a more accurately rendered RT3DE image is obtained (Figure 5A).

Cropping describes the process of manually moving a cutting plane from outside the 3D volume towards its center, thus providing a view from the cutting plane into the 3D volume.

Cropping is performed following the orthogonal acquisition planes (superior–inferior, anterior–posterior, left–right) or using a free plane.

We prefer a semi-automated method based on a crop box positioned in the region of interest. In our case, the MV apparatus includes mitral commissures and an entire excursion of the leaflets avoiding the dropout of echoes throughout the cardiac cycle (Figure 5B).

After obtaining the RT3DE volumetric cropped image, we can freely rotate the image 360 degrees in the best perspective (atrial view, ventricular view, etc.) (Figure 5C,D).

Usually, in the atrial view perspective, the aortic valve (AV) is displayed in an upper position [1,2].

The 3D reconstruction of the regurgitant jet can also be performed using the cropping mode, improving the detection of MVR jets origin.

## 3. Know Why

The introduction of 3DE has substantially improved the knowledge of anatomical and functional mechanisms of MV disease.

### 3.1. Anatomical Characterization

The RT3DE assessment of MV allows a better MV disease understanding than two-dimensional echocardiography (2DE). 3D-TEE has been shown to be more useful than 2D-TEE in assessing mitral valve prolapse (MVP) [16].

In addition, RT3DE allows us to visualize the whole MV.

Even though MV diseases can be easily studied by 2DE, morphological RT3DE imaging provides a precise characterization, thus facilitating diagnosis and accurately showing anatomical functional features. The possibility of choosing the best perspective for each cardiac structure results in a detailed anatomical characterization useful for planning a surgical or interventional strategy in MV diseases.

Preload conditions, as well as LV size and function, make MV evaluation challenging, particularly when a secondary etiology occurs [17,18].

Effective regurgitant orifice area (EROA) and regurgitant volume (RVol), as measured by proximal isovelocity surface area (PISA), may both overestimate and underestimate the severity of MR [18]. LV volume quantification is crucial [19], particularly before and after surgical or percutaneous correction. The ratio of mitral regurgitant volume (RVol) to left ventricular (LV) end-diastolic volume (EDV) (RVol/EDV) has been shown to be an independent prognostic factor in patients with secondary MR [20]. 2D-LV ejection fraction (LVEF) is the most commonly used parameter to determine cardiac function influencing VHD strategy. However, it does not allow evaluation of myocardial mechanics. On the other hand, several studies have shown that global longitudinal strain (GLS) is an independent predictor of mortality in patients with severe primary MR and permits unmasking subclinical LV dysfunction despite normal LVEF [5,6,7]. In secondary MR, GLS has been shown to be more accurate than LVEF in evaluating cardiac function. A more impaired GLS has been reported in patients with severe secondary MR compared with patients without MR with comparable LVEF values [21]. Moreover, GLS has been considered more sensitive in detecting any alteration in long-axis shortening. Longitudinal strain may identify a subclinical cardiac failure [22]. In addition, LV volume assessment by RT3DE has been demonstrated to be accurate, reproducible, and superior to conventional 2D methods. The superiority of the RT3DE approach has been demonstrated in various clinical scenarios, but its use is limited in patients with a poor acoustic window. LV volume and mass values obtained by RT3DE echocardiography are comparable with those derived from cardiac magnetic resonance imaging (MRI) or radioisotope techniques [23,24,25]. It has been shown that 3D-TEE is more accurate than 2D-TTE in identifying the degenerative etiology of M.R. such as Barlow’s disease (BD) and fibroelastic deficiency (FED) [12,26,27,28,29,30].

Visualizing the entire surface of the leaflets allows an easy identification and classification of the prolapsing, flail, and tethered segments [31]. Systematic use of the angle views to simplify the detection of the scallops has also been proposed, thus increasing the diagnostic accuracy in complex MV prolapse, such as commissural location [32,33]. These angle views, based on AV position, distinguish: (1) an anterior view with AV at 12 o’clock (surgical view or enface) to better analyze the anterior leaflet and its scallops (A1, A2, A3) Figure 6A; (2) an anterolateral view with AV in a lateral position to better analyze anterolateral commissure (ALC) and the near scallops (A1–P1) Figure 6B; (3) a posteromedial view with A.V. in medial position to better analyze posteromedial commissure (PMC) and the near scallops (A3–P3) Figure 6C; and (4) a posterior view with A.V. at six o’clock to better analyze the posterior scallops (P1, P2, P3) (Figure 6D).

The utility of the full volume RT3DE analysis in complex MVP has been well assessed [34,35].

The angle views (lateral and medial) help to detect commissural prolapse and its complications. The leaflet image’s motion flow and the realistic and dynamic vision of RT3DE imaging allows a “segment-oriented approach” method of analysis with significant implications in surgical planning (Figure 7A–D). It has also been demonstrated that RT3DE imaging was more accurate and reproducible than 2D imaging for assessing MV lesions, especially for evaluating complex multi-segment MV prolapse involving one or both leaflets compared with simple mono-leaflet lesions [36,37,38]. Therefore, RT3DE imaging allows the detection of atypical anatomies such as multiple middle posterior scallops composed of two or three subscallops or subclefts of the leaflets (Figure 7B). Therefore, the RT3DE anatomical diagnosis, especially in complex MV prolapse, provides the surgeon with precious information and has a very significant impact on surgical strategy. Recent guidelines on valve heart disease (VHD) recommend using RT3DE for primary mitral insufficiency studies [39]. Moreover, the role of RT3DE in congenital MV diseases (such as cleft or parachutes MV) is promising.

According to our experience, in patients with endocarditis RT3DE anatomical imaging accurately detects vegetation characteristics, such as the site, the extension, and the relationship with near structures (Figure 8A–F).

2DE may permit diagnosis of a traumatic or ischemic papillary muscle rupture. At the same time, RT3DE may be useful in differentiating the papillary muscle from its tip anchored to the leaflet prolapsing in the LA (Figure 9A–C). Furthermore, RT3DE anatomical characterization is successfully used in mitral stenosis (MS) for measuring MV area by planimetry.

### 3.2. Quantification of the Morphology

Visualizing the MV apparatus by RT3DE using specific software allows a deep understanding of the mechanisms of transvalvular regurgitation. Assessing valve morphology, geometry of the annulus, shape and surface of the leaflets, and the relationship between MV, the papillary muscle, and LV is useful for evaluating a surgical or percutaneous MV repair strategy. The morphological and semiautomatic assessment of MV apparatus improves the accuracy of diagnosis even when performed by an unskilled sonographer. Notably, identifying a closure line and the relative position of the papillary muscle tips provides a quantitative analysis of the MV anatomy before and after MV surgery and percutaneous interventions. LV measurements such as tethering distance and tenting volume are also helpful in decision making.

MVN analysis is used for distinguishing BD from FED [40]. It has been suggested a diagnostic algorithm based on the cut-off value of the height of prolapse (>1 mm) in order to recognize a degenerative MR etiology. Moreover, a cut-off value of prolapse volume (1.15 mL) has been proposed to evaluate the degenerative etiology [40]. Even though 2D imaging allows us to identify characteristic findings of both etiologies, a complete assessment of the MV annulus is only possible using MV navigation (Figure 10A,B). Both pathologies are characterized by annular dilatation and require an annuloplasty; however, in BD, the annular saddle shape and systolic displacement during coaptation are lost as measured by the annulus height/intercommissural diameter ratio. These features might influence the choice between the repair technique and annuloplasty ring, leading to a preference for a rigid and saddle-shaped annulus for BD and a flexible ring in FED [41]. The annulus’s dimensions, shape, and function, well studied using only MVN/MVQ, are essential for choosing a surgical technique.

### 3.3. Quantification of Mitral Regurgitation

Both semi-quantitative and quantitative methods for MR grading have been recommended [39,42,43,44,45]. Quantitative 2D color methods, such as the vena contracta area (VCA) measurement or the PISA, are useful for evaluating the severity of MR by calculating the effective regurgitant orifice (ERO) and regurgitation volume (RV). These planar measures have some disadvantages, especially in the case of eccentric and multiple jets and when the geometry of the ERO is complex. Geometric limitations of 2D echocardiography for quantifying MR severity could be overcome by RT3DE color. RT3DE color imaging allows the assessment of the volume, origin, and extension of the regurgitant jet and the relationship with adjacent structures. ERO RT3DE calculated using RT3DE-VCA and RT3DE-PISA was not inferior to the measurement obtained by MRI, which is the gold standard for quantifying LV stroke volume and regurgitant volume [46]. RT3DE-VCA is feasible using only the full volume acquisition. Multiplanar reconstruction is performed by positioning two orthogonal image planes to the regurgitant in order to visualize the jet origin (usually in mid-systole). Subsequently, a third plane perpendicular to the selected planes is moved along the length of the jet until its narrowest diameter is encountered, obtaining RT3DE-VCA. RT3DE-VCA is manually measured by planimetry of the color Doppler flow signal along the highest velocity aliased core corresponding to the value of ERO RT3DE (Figure 11). The estimation of MR is obtainable by multiplying RT3DE-VCA and the velocity–time integral of the regurgitant flow by continuous wave Doppler [47]. RT3DE PISA is obtained using software specifically developed for RT3DE-PISA determination (eSie PISA Volume Analysis; Siemens Medical Solutions Inc). Color Doppler aliasing velocities are set for 2D acquisition (37 ± 3 cm/s; range, 31–42 cm/s). Some software can perform a semi-automated quantification of 3D-PISA by visualizing a green cloud on RT3DE color Doppler images so that the aliasing velocity and initial seed point (an approximate location in RT3DE space that the software needs as an input to find the RT3DE flow convergence) can be selected for RT3DE-PISA analysis [48] (Figure 12). Therefore, the software fragment RT3DE volume data, applying an optimized segmentation algorithm [49], allows calculation of more than one PISA volume simultaneously if multiple jets are present. In the case of multiple jets, we should calculate the greatest PISA or VCA. However, according to the latest recommendation, it is advisable to use semiquantitative parameters for assessing MR severity, such as the MV velocity time integral (MVVTI)/LV outflow tract velocity time integral (LVOT VTI) ratio (>1.5). Recently, it has been suggested to calculate the regurgitation fraction (RF) obtained by adding the LV total stroke volume (SV) (end-diastolic volume minus end-systolic volume), effective SV, and regurgitant volume (RV) measured by 2D PISA. These measurements should be calculated in a phase of hemodynamic compensation [25]

RT3DE is also helpful for assessing ERO with the VCA method [50,51,52].

### 3.4. Quantification of MV Stenosis

Even though mitral calcifications (the main pathological finding of rheumatic and senile degenerative MV stenosis) are not detected by RT3DE due to its lower spatial resolution, the MPR modality can be used to more precisely measure the non-planar MV orifice at the tips of the leaflets compared with 2DE (Figure 13) [52].

## 4. Know When

RT3DE is useful in surgical or clinical MV decision making. The use of a checklist in order to improve RT3DE study should be encouraged [9,53].

### 4.1. Mitral Regurgitation

According to Carpentier’s classification, based on leaflet mobility anatomy and etiology (primary and secondary M.R), three types of MR have been proposed:

**Type I (normal leaflet motion with normal or dilated annulus):** this type includes secondary M.R. with dilated annulus due to LV dysfunction and remodeling and primary M.R. caused by congenital (cleft) or acquired (endocarditis) leaflet perforation. Typically, in primary MR the annulus is normal.

Even though secondary MR often results in a central regurgitation jet, eccentric and multiple regurgitation jets may occur. In these cases, RT3DE color Doppler is more helpful than 2DE in identifying and quantifying ERO. An RT3DE color cropped LV view may be more useful than the LA view in detecting the flow convergence regions of a regurgitant jet using a power color map (Figure 14). These data are crucial in evaluating the feasibility of MV repair and deciding the number and location of percutaneous clips.

**Type II (excessive leaflet motion):** this type includes both primary and secondary MR. In primary MR, MV prolapse has been described. MV is thought to be caused by BD and FED complications such as chordal rupture and flail. 2DE usually permits us to distinguish BD from FED. However, the 3D anatomical characterization of MV morphology overcomes the limits of 2DE in identifying commissural pericommissural scallop prolapse (A1–P1, A3–P3) or a BD with complex anatomy (Figure 15). Characteristically, leaflets in BD appear to be exceptionally soft and very redundant. Often, RT3DE allows the visualization of small clefts associated with the BD. In the FED diagnosis, the role of 3D-TEE is more limited. In FED, the leaflets are typically thinner and the annulus diameter is normal. The sclerosis of the leaflets occurs more frequently than in BD and is well shown by RT3DE. Typically, scallop P2 is involved, whereas commissural scallops are rarely interested. The role of RT3DE in differentiating the two etiologies is poor. MVQ and MVN are unique in showing structures otherwise understudied, such as the annulus and its functional geometry. Lastly, RT3DE color is very useful for evaluating eccentric jets (commissural) with non-planar ERO.

**Type IIIa (systolic and diastolic restricted motion):** this type includes primary MR due to degenerative calcifications of the annulus extended to the leaflets and a rheumatic etiology. 2DE has a higher accuracy than RT3DE for detecting mitral calcifications due to the better spatial resolution. RT3DE color is useful in detecting non-planar ERO (Figure 16A–D).

**Type IIIb (systolic restricted/tethered motion):** this type includes secondary MR caused by LV dilatation leading to papillary muscle displacement and symmetrical or asymmetrical leaflet tethering (central regurgitation or eccentric regurgitation 3DE with MVN or MVQ may be useful for planning MV repair (Figure 17).

### 4.2. Mitral Valve Stenosis

Rheumatic disease, progressive senile atheromatous degeneration of the mitral annulus, and leaflets in rheumatic/calcific forms have been reported to be mainly involved in the development of acquired MS [15,54,55]. MV replacement with mechanical or bioprosthetic valves is feasible in both cases. In the presence of well-established characteristics, percutaneous mitral balloon valvuloplasty (PMBV) could be considered for rheumatic MS treatment [56,57].

Echocardiography has a pivotal role in the choice of the best treatment based on morphological and functional severity assessment. Most recent guidelines recommend 2DE for the complete evaluation of MS, even though several limitations must be considered due to the non-planarity of the stenotic orifice and reverberations of calcifications assessing MV planimetry may be difficult. Furthermore, Doppler echocardiographic measurements, including the continuity equation, pressure half-time (PHT), and PISA for calculating functional MV area, have several limitations [56,57] that might be overcome using RT3DE.

Both longitudinal parasternal and apical views have been proposed to obtain the ventricular perspective of the MV apparatus. In our experience, the apical view allows a better ventricular visualization of the stenotic mitral orifice. Remarkably, a multiplanar reconstruction of MVA with a better alignment of the image plane at the mitral tips provides a more accurate and reproducible planimetric measurement [58,59]. Three-dimensional planimetry is nearer to Gorlin’s equation planimetry than 2D planimetry. The semiautomatic assessment of RT3DE orifice area (3DOA) by MVN has been shown to be more effective in detecting severe MS than RT3DE planimetry and Doppler quantification [53,60].

The MV area obtained with RT3DE PISA has also been compared with RT3DE and 2DE planimetry. The semi-automated RT3DE quantification of the MV area might be more accurate than conventional 2D methods [61,62].

RT3DE is used, together with 2DE, to evaluate the potential PMBV feasibility and to predict the success. An RT3DE-based score for PMBV has been developed to improve the prediction of success of the 2DE-based Wilkins score [63].

The severity of valvular calcification and its extension is the stronger predictor of adverse events after PMBV [42,64,65,66,67].

RT3DE planimetry is more accurate than 2D planimetry in identifying the non-planar orientation of MV leaflets. Moreover, the true MV area at the tip of the leaflets can be measured (Figure 18A,B).

Conversely, the low spatial resolution of the RT3DE image does not easily allow us to differentiate calcium from valvular tissue. Therefore, we suggest using RT3DE only when the 2DE study is not useful for evaluating the MS severity and before PMBV.

### 4.3. Mitral Valve Repair Procedures

The 3D-TEE detection of echographic features is very useful in evaluating percutaneous or surgical repair and accurately assessing MV, LA, interatrial septum, and catheter moving segments.

### 4.4. Surgical Repair

A complete understanding of the underlying degenerative etiology, anatomical lesions, and leaflet or annulus dysfunction is mandatory to predict the efficacy of repair and to choose the most appropriate repair technique.

Planning leaflet resection, evaluating the type and sizing of prosthetic rings, and avoiding systolic anterior motion (SAM) are the three goals to achieve [68].

On the other hand, only a few pieces of data underline the importance of 3DE during a surgical procedure. Conversely, MV repair failure repair may be detected early using RT3DE (Figure 19).

### 4.5. Percutaneous Repair

An anatomical assessment of the MV apparatus is needed before percutaneous repair. The EVEREST [69,70] echocardiographic criteria include: non-rheumatic/endocarditic valve morphology, ≥4 cm^2^ MV area, sufficient leaflet tissue, flail gap ≤ 10 mm, flail width ≤ 15 mm, coaptation depth ≤ 11 mm, and coaptation length ≥ 2 mm. However, it has been shown that complex MV lesions can be successfully treated with MitraClip [71,72,73,74] surgery. Intraprocedural guidance of MitraClip repair is based on 2D- and 3D-TEE imaging [75,76,77,78,79].

The initial step is the guidance of the transseptal puncture. The ideal location for the puncture in most patients is the mid-fossa in the bicaval view, and the puncture site is directly visualized by the ‘tenting’ of the fossa ovalis. 3D-TEE with X-plane modality provides imaging of the entire interatrial septum (IAS). It allows the correct position and orientation of the Mullins Catheter, avoiding complications (catheter distance from aortic valve and atrial roof, Figure 20A,B).

2.The second step is the introduction of a steerable guide catheter (SGC) and advancement of the clip delivery system (CDS) into the L.A.; RT3DE with X-plane modality permits us to avoid LA free-wall injury (Figure 20C).3.The third step is positioning the Mitraclip above the MV. RT3DE with zoom modality en face view allows determination of when the clip is adequately positioned above the middle segments of the MV and if the orientation is perpendicular to the line of coaptation (Figure 20D).4.The fourth step is the advancement of the Mitraclip in the LV. RT3DE with zoom modality in the L.A. or LV views represent direct visualization of the MitraClip concerning the MV and the line of coaptation, lowering the overall gain to get a clear impression of the clip in the LV views (Figure 20E).5.The fifth step is grasping the leaflets and assessing proper leaflet insertion and clip detachment.

2DE is the preferred method for monitoring the procedure in this crucial step because of the low frame rate of RT3DE imaging [58]. This is true for clips released in the central A2–P2 position.

However, many clips may be grasped in a non-central position, so the classic bi-dimensional three-chamber view is not appropriate, and the RT3DE X-plane modality provides accurate localization of the grasping view by placing the interrogation line through the clip in the commissural view and visualizing tissue grasp in the orthogonal view.

6.The sixth and final step is assessing the procedure results. RT3DE color may be used to evaluate the residual MV regurgitation, and 3D-TEE multiplanar reconstruction may be used to assess the post-implant mitral valve area. The clip can be repositioned or withdrawn in case of severe mitral stenosis. In case of residual significant MV regurgitation, another clip can be released.

3D-TEE is advantageous in the majority of the steps for the MitraClip procedure [80], including the guidance of the clip delivery system within the left atrium (LA), visualization of the guide length within the LA, positioning of the clip in the appropriate location, and alignment of the opened clip relative to the commissures. 2D-TEE was superior in visualizing leaflet grasping. Using 2D- and 3D-TEE is associated with a remarkable 28% reduction in procedure times [81].

It has been hypothesized that 3DE is useful not only in surgical or clinical decision making but also in the short-term management of MVD due to the fact that it detects most complications early, such as detachments of the prostheses due to endocarditis or pannus or prosthesis thrombosis. Moreover, a role for RT3DE in the long-term management of MVD has been proposed [82].

## 5. Limitations and Future Perspectives

The main limitations of RT3DE are related to physicians’ expertise, considering that certificated training is required.

In this paper, we highlighted the “tip and tricks“ of the RT3DE acquisition modalities and post-processing to provide a practical guide for the echocardiographer to facilitate and accelerate the learning process.

Several advantages of RT3DE, such as allowing simultaneous visualization of more than one cardiac structure, which enhances the procedure’s safety; lower procedural complications; and shorter interventional times, have been reported.

However, incorporating RT3DE into the workflow may be cost effective and useful in clinical practice [83]. Currently, RT3DE has not been strongly included in the latest valvular heart disease (VHD) recommendations or guidelines. “Three-dimensional TEE may be helpful in further visualizing the abnormal mitral valve anatomy, offering a “surgical” view of the valve” is stated in the latest ACC/AHA Guidelenes [39]. Therefore, its role is deemed to be more important in the future.

## 6. Conclusions

Important advances in RT3DE imaging have recently been reported. The introduction of RT3DE in clinical practice using both transthoracic and transesophageal modalities has improved spatial image resolution and anatomical characterization of the cardiac structures, allowing a real-time visualization of the valves and subvalvular apparatus, volume quantification, surgical timing, intraprocedural guidance, and post-procedural evaluation.

## Figures and Tables

**Figure 1 jcm-12-02522-f001:**
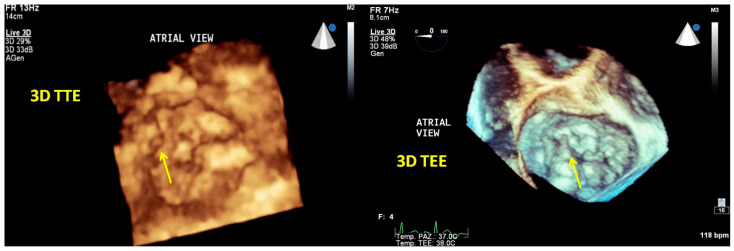
Tridimensional transthoracic echocardiography (3D-TTE) quality (**left panel**) compared with three-dimensional transesophageal echocardiography (3D-TEE) (**right panel**). Both images show an LA view of MV prolapse with a P1 flail (yellow arrow) in a patient with Barlow disease. 3D-TEE allows a more refined representation of the morphological alterations and the ruptured chords compared with 3D-TTE.

**Figure 2 jcm-12-02522-f002:**
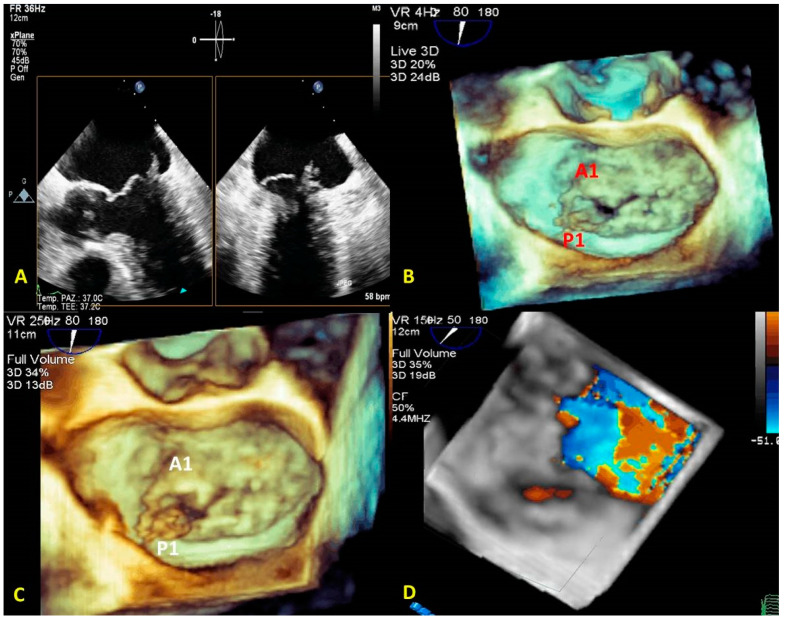
MV endocarditis in a patient with Barlow disease. (**A**) X-plane modality simultaneously shows the long axis and the commissural views: vegetation on the posterior leaflet is visualized. (**B**) The 3D-zoom modality allows the identification of two different vegetations on A1 and P1 scallops. (**C**) Full volume modality allows a more detailed anatomical characterization of filamentous-like vegetation on scallop A1 and of the polypoid vegetation on scallop P1. (**D**) Color 3D modality shows a main regurgitation jet originating from A1-P1 scallops and a minor jet originating from the P2 scallop. MV: mitral valve.

**Figure 3 jcm-12-02522-f003:**
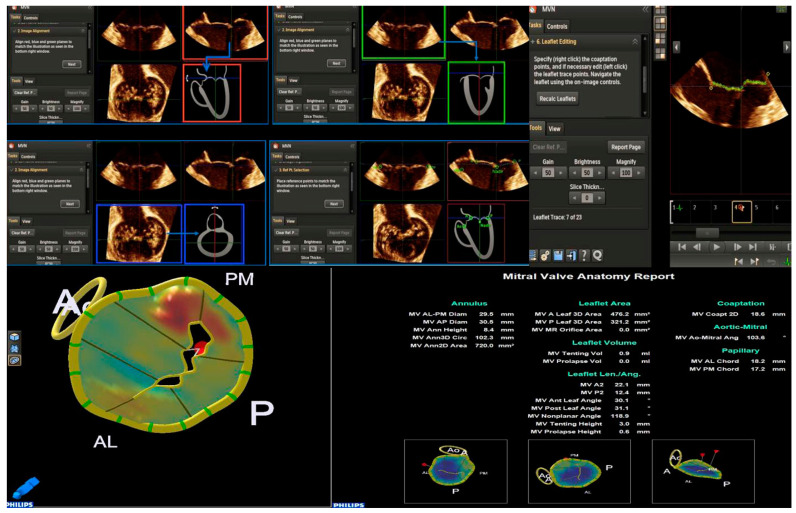
The control panel of the MVN is shown. The various parameters obtained by MVN are displayed on the bottom right. Each step is explained in the text. MVN: mitral valve navigation.

**Figure 4 jcm-12-02522-f004:**
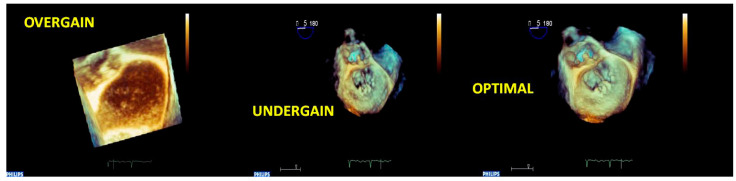
Acute endocarditis of MV in a patient with a previous surgical mitral repair. The sequence of 3D-zoomed images acquired with different gain settings is shown. The left image is noisy because of the overgained acquisition; the central image offers a low resolution of the cardiac structures because of the undergained acquisition; the right image is the optimal one, and the vegetation on the mitral annuloplasty suture is well assessed using the optimal gain setting. MV: mitral valve.

**Figure 5 jcm-12-02522-f005:**
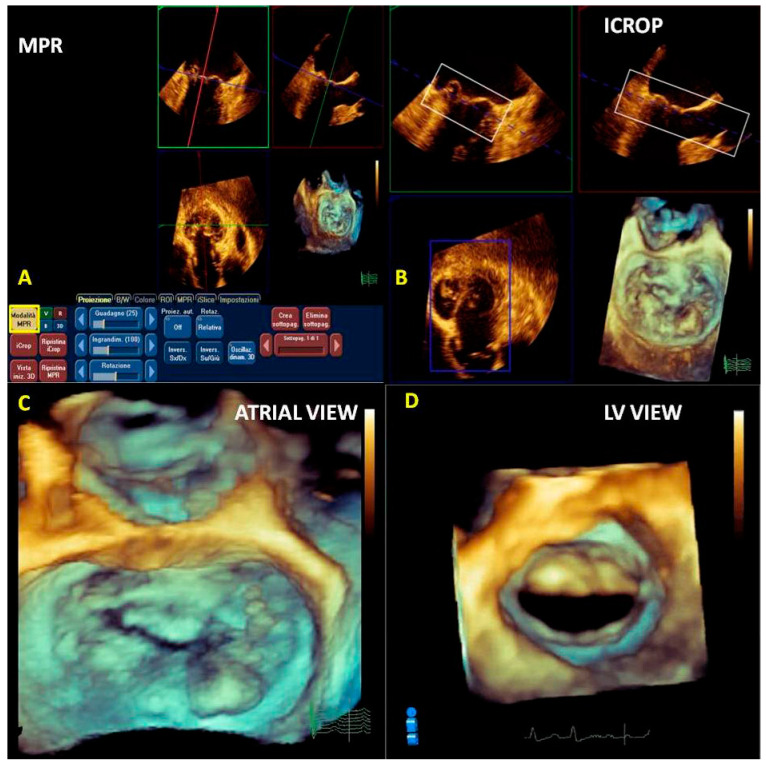
**A** 3D full volume acquisition of a P3 prolapse and chordal rupture of PMC. (**A**) MPR: longitudinal plane (red line), transversal plane (blue line), and sagittal plane are positioned perpendicularly at the region of interest (prolapse). (**B**) Icrop modality: after the correct alignment of the planes, the box crop is moved toward the region of interest, and the resulting image is displayed on the bottom right. (**C**) The 3D image obtained can be rotated to obtain the best perspective (360°), such as in an LA view or (**D**) in an LV view. PMC: posteromedial commissure; MPR: multiplanar reconstruction; LA: left atrial; LV: left ventricular.

**Figure 6 jcm-12-02522-f006:**
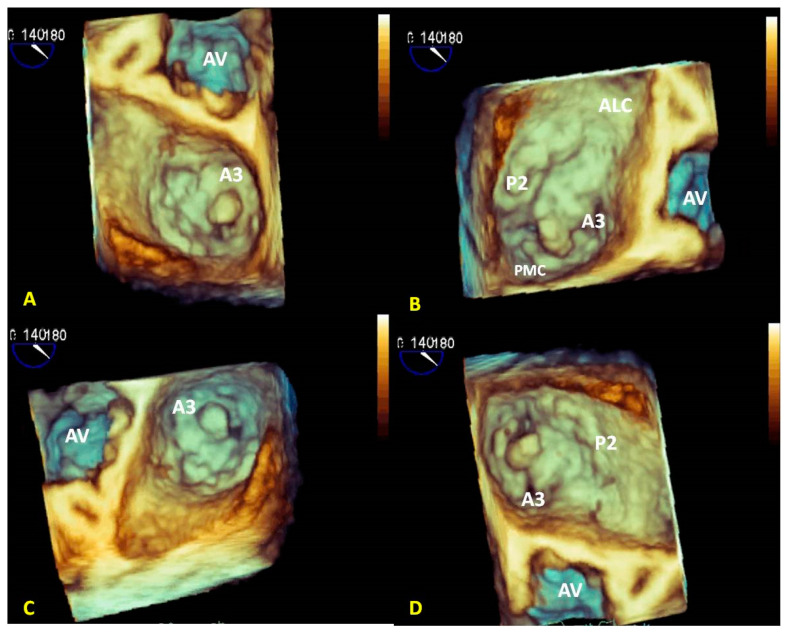
Cropped angled views of A3 prolapse due to chordal rupture of the PMC. (**A**) AV in the upper position: this view allows the optimal identification of the anterior leaflet. (**B**) AV in lateral position: this view allows the optimal analysis of AMC. (**C**) AV in the medial position: this view allows the optimal analysis of PMC. (**D**) AV in a lower position allows the optimal analysis of the posterior leaflet. The (**B**–**D**) images clearly detect the rupture of the chordae of the PMC but not the (**A**) image. PMC: posteromedial commissure; AV: aortic valve; AMC: anterolateral commissure.

**Figure 7 jcm-12-02522-f007:**
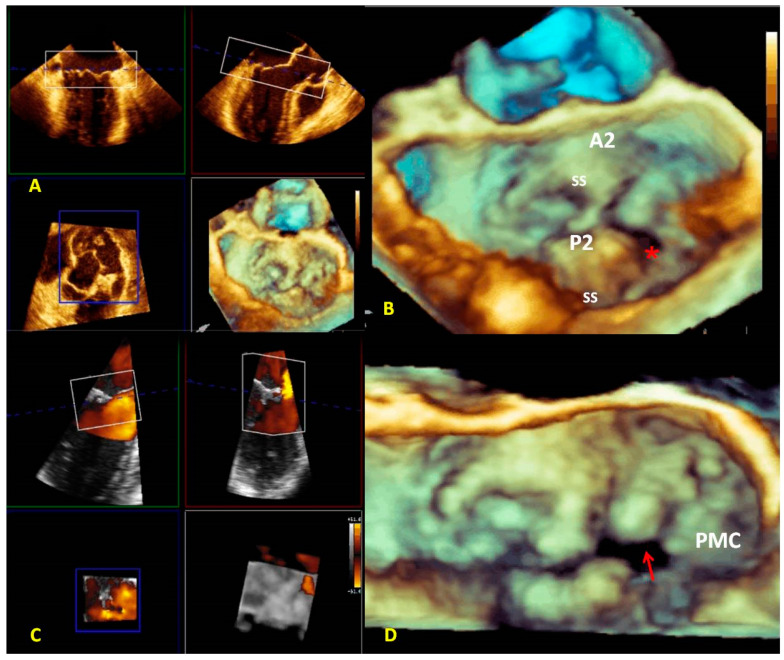
A case of complex Barlow disease. (**A**) MPR and cropped image of A2–P2 prolapse. (**B**) LA view of the RT3DE rendered image after the cropping reveals the partition of the scallops A2 and P2 in subscallops and a cleft (*) between P2 and P3. (**C**) RT3DE color Doppler allows the identification of the eccentric regurgitant jet at the level of the PMC. (**D**) A more refined RT3DE rendered image permits clear identification of a cleft between P2 and P3 (arrow) and clearly shows the PMC prolapse. This figure shows how the different RT3DE modalities can add different diagnostic information. MPR: multiplanar reconstruction; PMC: posteromedial commissure.

**Figure 8 jcm-12-02522-f008:**
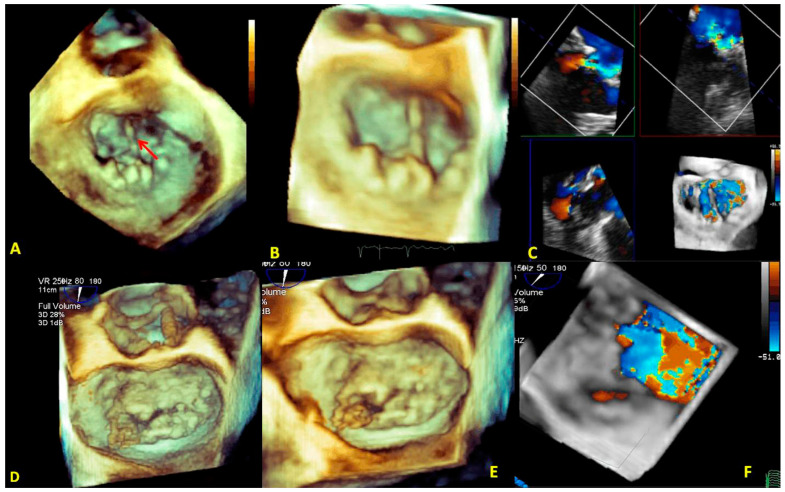
Panel with figures (**A**–**C**) represents the power of RT3DE anatomical characterization (**A**,**C**) to localize the vegetation (arrow) and its pathological repercussion (RT3DE color strongly underlines the eccentric jet of regurgitation). Panel with figures (**D**–**F**) shows the ability of 3DAC to refine the form, the size, and the position of the vegetation. 3DAC: three-dimensional anatomical characterization.

**Figure 9 jcm-12-02522-f009:**
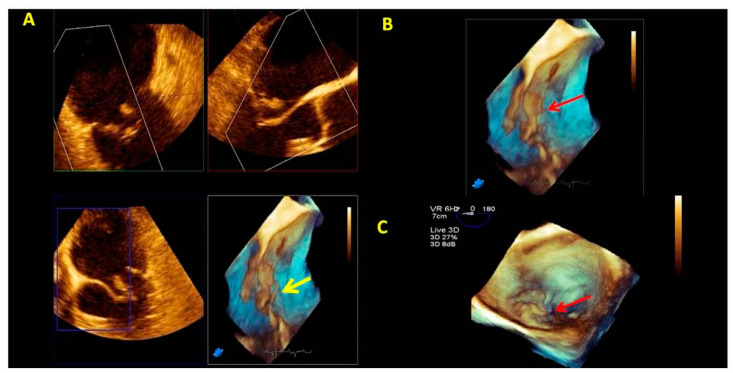
A case of MV papillary rupture complication of myocardial infarction. (**A**) MPR and very eccentric RT3DE cropped images depict a ruptured chord (yellow arrow). (**B**) RT3DE cropped images allow distinguishing of the ruptured head of the papillary muscle hanging on the posterior MV leaflet and prolapsing in the left atrium (red arrow). (**C**) A further ruptured chord can be seen in the left atrium (red arrow). MV: mitral valve; MPR: multiplanar reconstruction. MV: mitral valve; MPR: multiplanar reconstruction.

**Figure 10 jcm-12-02522-f010:**
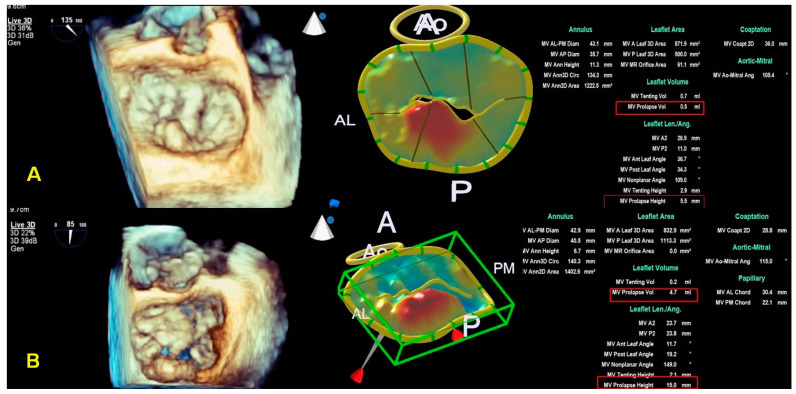
Panel (**A**), a case of FED. Panel (**B**), a case of BD. The two panels represent the typical report of MVN for the two different etiologies. The height and volume of MV prolapse are the main MVN parameters differentiating the etiologies. See the text for the cut-off values. FED: fibroelastic deficiency; BD: Barlow disease; MVN: mitral valve navigation; MV: mitral valve.

**Figure 11 jcm-12-02522-f011:**
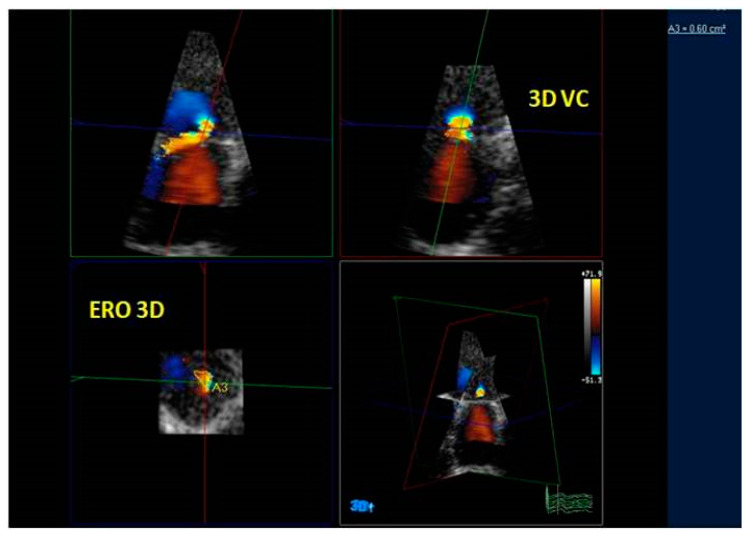
A case of eccentric MV regurgitation. The 3D TTE color Doppler allows the estimation of the ERO and RV by the measurement of the 3D-VCA. 3D-TTE: three-dimensional transthoracic echocardiography; ERO: effective regurgitant orifice; VR: regurgitation volume; 3D-VCA: three-dimensional vena contracta area. 3D-PISA: three-dimensional-proximal isovelocity surface area.

**Figure 12 jcm-12-02522-f012:**
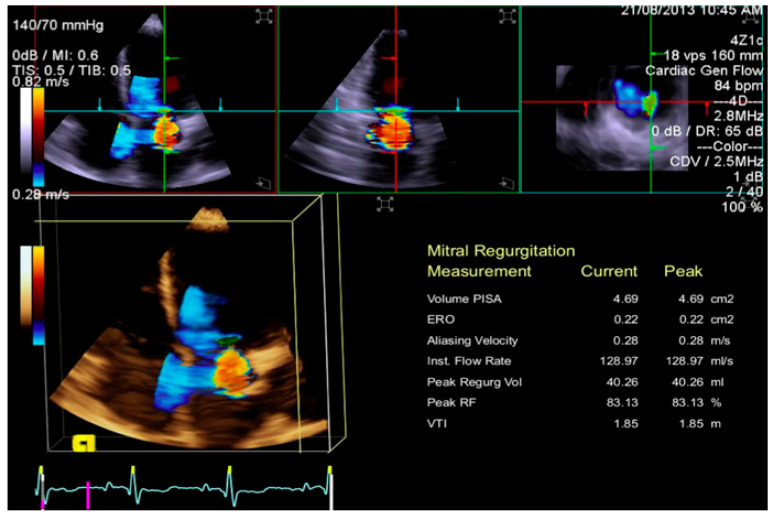
3D-TTE of a moderate MV regurgitation. By moving the planes perpendicularly to the tips of MV, MPR allows the reconstruction of the regurgitant mitral orifice, and its area can be measured with more precision 3D-TTE: three-dimensional transthoracic echocardiography; MPR: multiplanar reconstruction. ERO = Effective Regurgitant Orifice.

**Figure 13 jcm-12-02522-f013:**
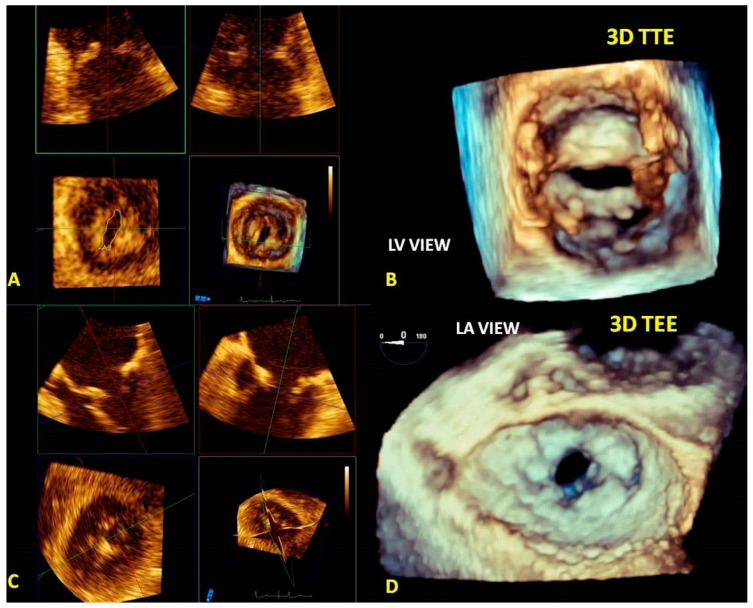
(**A**–**D**). 3D-TTE of a mild MV stenosis. (**B**) 3D-TEE of a severe MV stenosis. (**B**–**D**). By moving the planes perpendicularly to the tips of MV, multiplanar reconstruction (MPR) allows the reconstruction of the stenotic mitral orifice, and its area can be measured with more precision.

**Figure 14 jcm-12-02522-f014:**
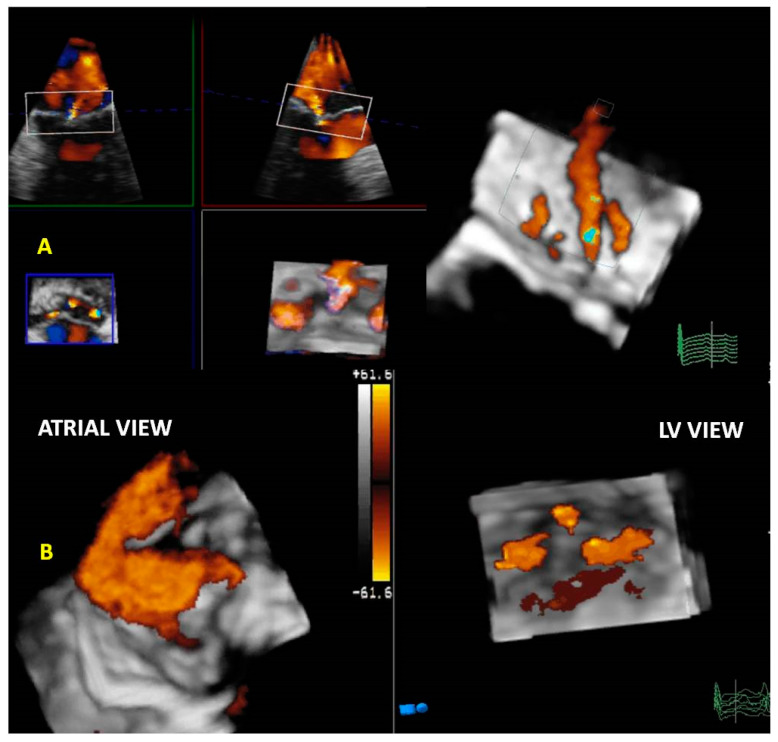
(**A**) 3D-TEE color cropping in a case of mitral regurgitation with multiple jets. In the MPR modality, the jets are preliminarily intersected perpendicularly and are successively cropped (a white box in the figure is placed on the jets). On the top right, the RT3DE cropped image shows the multiple regurgitation jets (the main at the level of A2-P2) from LA view. (**B**) 3D-TEE power color map of multiple jet regurgitation. An LA view shows an eccentric jet on the bottom left. On the bottom right, an LV view shows a multiple jet regurgitation. The LV view allows locating the flow convergence (jet origin) with more precision. 3D-TEE: three-dimensional transesophageal echocardiography; LA: left atrial; LV: left ventricular; MPR: multiplanar reconstruction.

**Figure 15 jcm-12-02522-f015:**
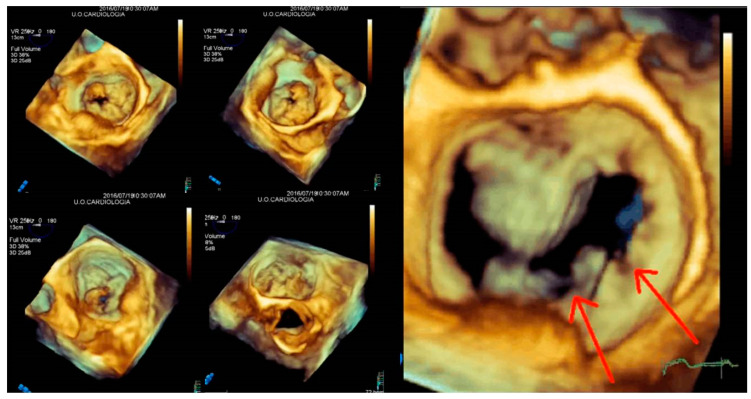
3D-TEE of BD with flail valve. On the left the panel are displayed different angled LA views showing the involvement of both leaflets; however, the anterior leaflet presents a more extensive myxomatous degeneration with respect to the posterior leaflet, and the A1 scallop flail can be detected. The right figure represents a zoomed RT3DE rendered image of two clefts of the posterior leaflet (red arrows). BD: Barlow disease; LA: left atrial.

**Figure 16 jcm-12-02522-f016:**
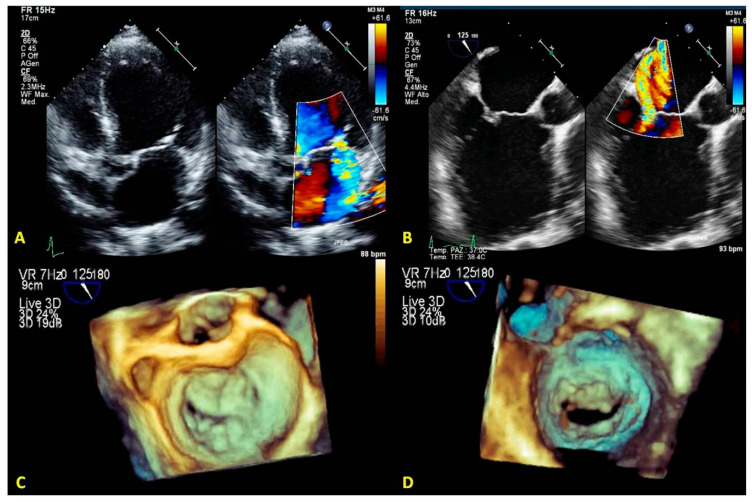
A case of type IIIa degenerative senile and TLMAC MV regurgitation. (**A**,**B**) 2D-TEE focused on mitral valve apparatus. Dual plane (left) shows fibrosis of leaflets and extensive annular calcification. Regurgitant jet originating from mitral leaflets (right) is depicted. (**C**,**D**): 3D-TEE reconstruction of the mitral valve. The left atrial view is represented in the classical surgeon’s view (left). Omogeneus posterior annular calcification appears as a protuberance undistinguishable from other structures; the low frame rate does not permit precise tissue characterization. A ventricular view is presented (right). Irregular calcifications involving subvalvular apparatus are distinguishable from other cardiac structures. 2D-TTE: two-dimensional trans-thoracic echocardiography; 2D-TEE: two-dimensional transesophageal echocardiography; LA: left atrial; LV: left ventricular; PMC: posteromedial commissure; ALC: anterolateral commissure; MV: mitral valve; TLMAC: tumor-like mitral annular calcification.

**Figure 17 jcm-12-02522-f017:**
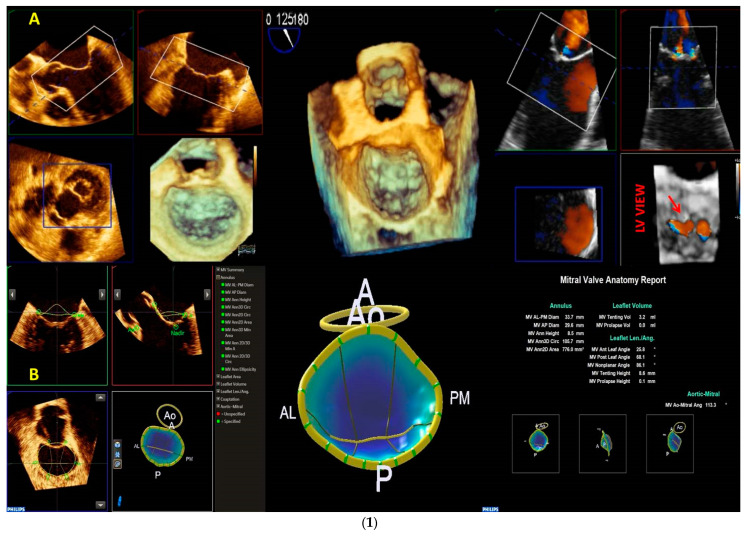
(**1**) Secondary MV regurgitation by symmetric tethering of both leaflets. (**Panel A**). MV cropping, MV rendered image, and cropped RT3DE color modality are shown. Multiple regurgitant jets are well detected by RT3DE color, and from an LV view, the precise localization of the flow convergence can be appreciated (A2-P2 and A3-P3). (**Panel B**) MVN of the MV. The tenting height (8.6 mm) is diagnostic of symmetric tethering, and its value is important for the suitability of percutaneous MV repair. In this patient, two clips were implanted. MV: mitral valve; LV: left ventricular; MVN: mitral valve navigation. (**2**) Ischemic MV regurgitation by tethering of the posterior leaflet (asymmetric regurgitation). (**Panel A**) 2D TTE, APICAL longitudinal VIEW for LVOT clearly shows the tethering of the posterior leaflet by the scarred LV posterior wall and the pseudo-prolapse of the anterior leaflet with the resultant eccentric jet (left). 2D-TTE, apical 2 chamber view, the inferior wall aneurysm is well defined and two regurgitant jets are detected (right). (**Panel B**) MVN obtained by 3D-TEE does not show an important tenting height. MV: mitral valve; TTE: trans-thoracic echocardiography; LVOT: left ventricular outflow tract; MVN: mitral valve navigation; 3D-TEE: three-dimensional transesophageal echocardiography.

**Figure 18 jcm-12-02522-f018:**
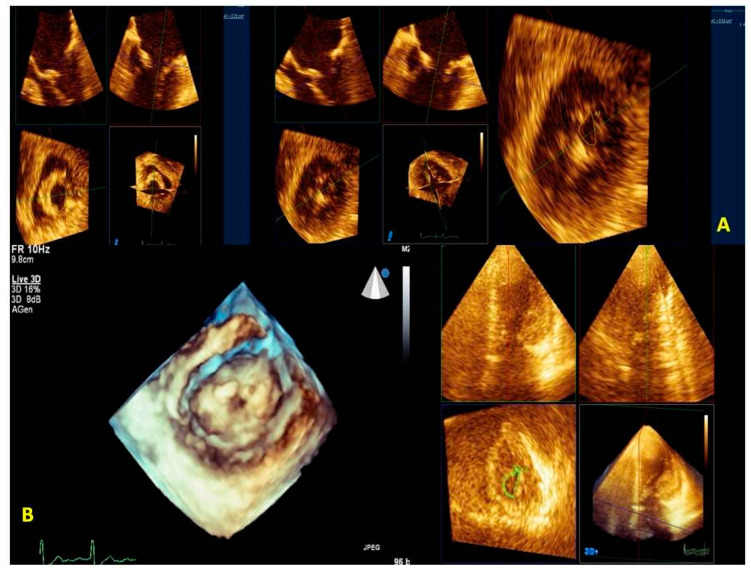
Mitral valve Stenosis. (**Panel A**) 3D-TEE. A greater MV area is measured for the incorrect position of the blue plane placed on the plane orthogonal to the mitral valve orifice (left). In the central figure, the correct position of the blue plane is shown. The correct position of the plane allows the proper representation of the MV orifice allowing the correct measurement of the planimetric area (right). (**Panel B**) 3D-TTE LV view of MV stenosis (left, Clip 5). The MPR of the MV area is performed at the level of the tips of the leaflets (right). Transthoracic imaging offers suboptimal imaging. MV: mitral valve; MPR: multiplanar reconstruction; 3D-TTE: three-dimensional transthoracic echocardiography; LV: left ventricular; 3D-TEE: three-dimensional transesophageal echocardiography.

**Figure 19 jcm-12-02522-f019:**
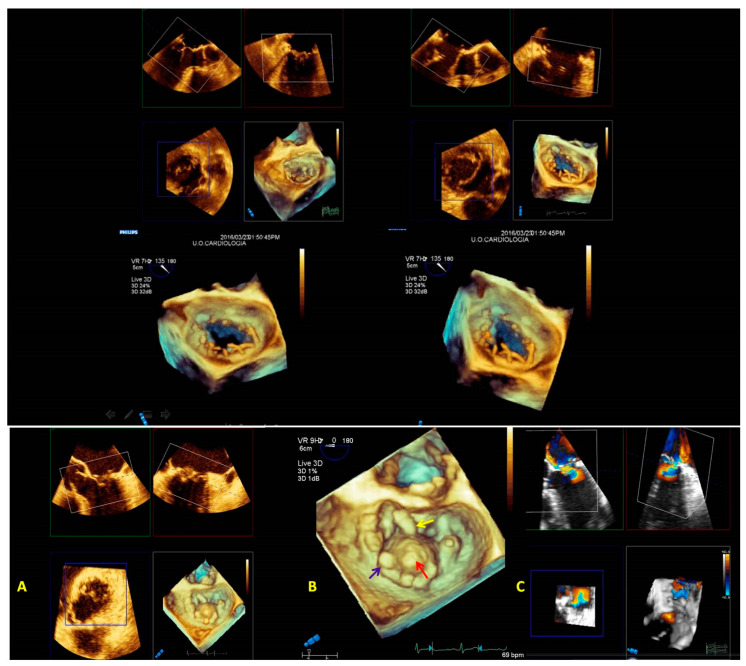
Surgical mitral valve repair performed via right mini-thoracotomy access (top). The upper part shows a 3D-TEE MPR with the cropping modality of a surgical MV repair; on the left is the systolic phase and on the right is the diastolic phase. On the lower part is shown RT3DE rendered images from the LA view. The 3D appearance of the prosthetic annular sutures is peculiar when MV repair is performed through the right mini-thoracotomy access. This is an example of a successful MV repair. 3D-TEE: three-dimensional trans-esophageal. echocardiography; MV: mitral valve; LA: left atrial; MPR: multiplanar reconstruction. A case of early endocarditis on MV repair performed on a Barlow disease patient (bottom). (**A**) A 3D-TEE MPR with cropping modality of an MV repair. (**B**) The RT3DE rendered image clearly depicts the prosthetic annular suture (blue arrow), the endocarditic vegetation on the suture (yellow arrow), and the consequently recurrent prolapse (red arrow). (**C**) The 3D color cropped image documents an eccentric regurgitant jet at the level of a detached suture caused by endocarditic vegetation. MV: mitral valve; 3D-TEE: three-dimensional transesophageal echocardiography; MPR: multiplanar reconstruction.

**Figure 20 jcm-12-02522-f020:**
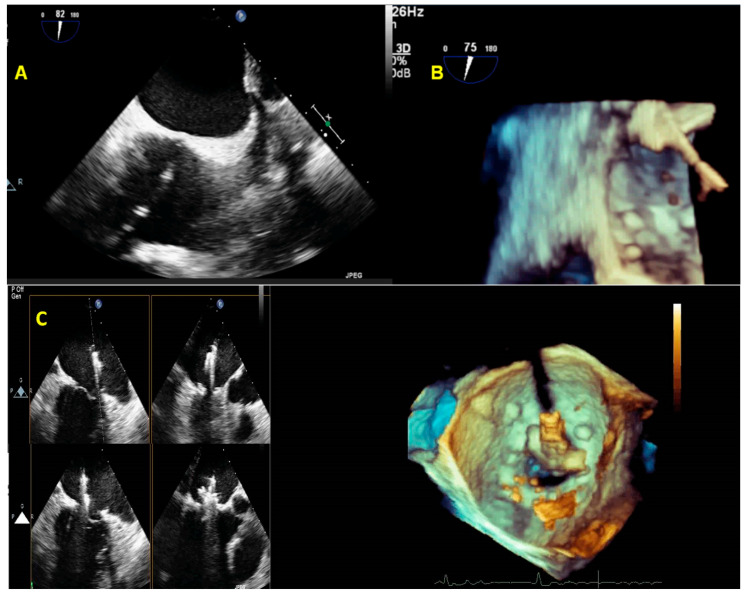
(**A**) 3D-TEE with X-plane modality permits us to orient the Mullins Catheter on the fossa ovalis. The tenting of the fossa ovalis is the marker for the correct position to perform the septal puncture; (**B**) RT3DE zoomed image allows precisely localizing the SGC in the LA. This is important to avoid LA free-wall injury. 3D-TEE: three-dimensional transesophageal echocardiography; SGC: steerable guide catether; LA: left atrium; (**C**) (**left**) A 3D-TEE with X-plane modality allows visualization of CDS in the LA. (**right**) A 3D zoomed real-time image allows for orienting the CDS above the middle segment of MV and perpendicularly to the MV coaptation line. 3D-TEE: three-dimensional transesophageal echocardiography; CDS: clip delivery system; MV: mitral valve. (**D**) On the left, X-plane modalities with and without color show the advancement of the Mitraclip in the LV. On the top right, an RT3DE zoomed image in the LA view allows us to directly visualize the Mitraclip in relation to MV coaptation line. On the bottom right, an RT3DE color image in the LV view with a reduced gain is obtained to clearly visualize the Mitraclip in the left ventricle (the asterisks show the double orifice. LA: left atrium; LV: left ventricle. (**E**) RT3DE modalities to evaluate the procedure result. On the top left, an X-plane modality with color (in commissural view) shows the grasping of the leaflets. The top right shows the mild MV gradient after the release of the clip. On the bottom left, the MPR modality shows a significant reduction in the MV regurgitation. On the bottom right, the RT3DE LV view shows the typical double orifice after the Mitraclip release. MV: mitral valve; MPR: multiplanar reconstruction; RT3DE: real-time three-dimensional echocardiography.

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
