# Peer review of "Clinical Utility of Three-Dimensional Echocardiography in the Evaluation of Mitral Valve Disease: Tips and Tricks"

_jcm, 2023, doi:10.3390/jcm12072522_

Round 1

Reviewer 1 Report

The authors provided a review of the usefulness of 3D- echocardiography in the evaluation of mitral valve anatomy and function with some practical tips and tricks. However, the manuscript needs an extensive re-organization, particularly of the technical section (“know-how”). In addition, the review will also further benefit of an extensive editing of English language and style.

Specific comments:

-       Table 1. Section 3D TTE – please substitute “acquisistion” with “acquisition” in both the lines. Section 3D TEE please clarify the angle provided (different from the angles provided in the main text lines 37-38).

-       An uniform style between the three tables will be more appropriate to ease reading of the paper.

-        Please consider eliminate Table 3 since it does not add any further relevant information.

-       The paper could benefit of a Limitation/challenges section as well as of a Future perspective section.

-       Do the authors think that volumetric assessment and deformation myocardial analysis provided by 3D echocardiography could have a role in evaluating mitral function? If yes, may add some relevant comments to the text.

Author Response

Reviewer 1

The authors provided a review of the usefulness of 3D- echocardiography in the evaluation of mitral valve anatomy and function with some practical tips and tricks. However, the manuscript needs an extensive re-organization, particularly of the technical section (“know-how”). In addition, the review will also further benefit of an extensive editing of English language and style.

ANSWER: We agree with the reviewer. The paper has been edited by a native English Teacher. Moreover, the review has been extensively re-organized accordingly

Specific comments:

-       Table 1. Section 3D TTE – please substitute “acquisistion” with “acquisition” in both the lines. Section 3D TEE please clarify the angle provided (different from the angles provided in the main text lines 37-38).

ANSWER: We corrected it accordingly.

-       An uniform style between the three tables will be more appropriate to ease reading of the paper.

ANSWER: We corrected it accordingly.

-        Please consider eliminate Table 3 since it does not add any further relevant information.

ANSWER: We agree with the reviewer and corrected it accordingly.

-       The paper could benefit of a Limitation/challenges section as well as of a Future perspective section.

ANSWER: We agree with the reviewer and added this section accordingly.See page 11, lines 541-552

-       Do the authors think that volumetric assessment and deformation myocardial analysis provided by 3D echocardiography could have a role in evaluating mitral function? If yes, may add some relevant comments to the text.

ANSWER: We thank the reviewer for suggesting this item. We think that volumetric assessment and deformation myocardial analysis provided by 3D echocardiography play a role in evaluating mitral function. We discussed this point. See page 5, lines 202-228.

Reviewer 2 Report

Interesting manuscript providing practical guidance on a signifant topic regarding 3DE in MV disease. Figures are of good quality with  educational character.

There are some points that should be addressed in order to raise the competence and strengthen the educational scope of the manuscript.

-Lines 37-39 the authors state  "3D-TEE is performed through the mid-oesophageal 37 view at 0 ° (4 chambers), 90° (2 chambers), and 120° (aortic valve long axis view) [8,9]; and 38 provides unique RT3DE images [10-12]. (Table 1)." However in table 1 they state "0°, 70°, 130° mid-esophageal views without color." This information is rather confusing and should be clarified.

-lines 74-76. It would be usefull to clarify why representation of Color Doppler is only possible with "Full Volume."

-Lines 290-299. The authors should further clarify the differences in morphology identified with 3DE between Barlow's and FED.

-Page 12. Ιn case of multiple  jets, how the MR severity should be quantified using VC and PISA method in authors' opinion?

-Is there a role for 3DE in the long term evaluation of MV disease after surgical or percutaneous treatment? Some practical guidance regarding  the indications, the parameters used and the limitations would be valuable.

Author Response

Reviewer 2

Interesting manuscript providing practical guidance on a signifant topic regarding 3DE in MV disease. Figures are of good quality with  educational character.

There are some points that should be addressed in order to raise the competence and strengthen the educational scope of the manuscript.

-Lines 37-39 the authors state  "3D-TEE is performed through the mid-oesophageal 37 view at 0 ° (4 chambers), 90° (2 chambers), and 120° (aortic valve long axis view) [8,9]; and 38 provides unique RT3DE images [10-12]. (Table 1)." However in table 1 they state "0°, 70°, 130° mid-esophageal views without color." This information is rather confusing and should be clarified.

ANSWER: We agree with the reviewer. We corrected it accordingly

-lines 74-76. It would be useful to clarify why representation of Color Doppler is only possible with "Full Volume."

ANSWER: We agree with the reviewer. We added this point. See pag2, lines 82-89

-Lines 290-299. The authors should further clarify the differences in morphology identified with 3DE between Barlow's and FED.

We agree with the reviewer. We added this point. See page 8, lines 384-390

ANSWER: -Page 12. Ιn case of multiple  jets, how the MR severity should be quantified using VC and PISA method in authors' opinion?

ANSWER: We agree with the reviewer. In the case of multiple jets, we should calculate the greatest PISA or VCA. We discussed this point. See page 7, lines 331-338

-Is there a role for 3DE in the long term evaluation of MV disease after surgical or percutaneous treatment? Some practical guidance regarding  the indications, the parameters used and the limitations would be valuable.

ANSWER: We agree with the reviewer. We discussed this point. See page 11, lines 535-540.

Reviewer 3 Report

I have a few complaints:

1) Introduction: I do not understand what does the shortcut ..M. … mean???

2) the font size does not match throughout the text: avoiding Systolic Anterior Motion (SAM)

3) minor punctuation errors

Author Response

Reviewer 3

I have a few complaints:

  • Introduction: I do not understand what does the shortcut ..M. … mean???

ANSWER: We agree with the reviewer. We corrected the mistake (MV: mitral valve)

  • the font size does not match throughout the text: avoiding Systolic Anterior Motion (SAM)

ANSWER: We agree with the reviewer. We corrected accordingly

  • minor punctuation errors.

ANSWER: We agree with the reviewer. We corrected accordingly

Round 2

Reviewer 1 Report

The overall quality of the manuscript improved after the revision process.